# Dynamic intramolecular regulation of the histone chaperone nucleoplasmin controls histone binding and release

Christopher Warren[1], Tsutomu Matsui[2], Jerome M. Karp[1], Takashi Onikubo[1,3], Sean Cahill[1], Michael Brenowitz[1], David Cowburn [1], Mark Girvin[1] & David Shechter[1]

Nucleoplasmin (Npm) is a highly conserved histone chaperone responsible for the maternal storage and zygotic release of histones H2A/H2B. Npm contains a pentameric N-terminal core domain and an intrinsically disordered C-terminal tail domain. Though intrinsically disordered regions are common among histone chaperones, their roles in histone binding and chaperoning remain unclear. Using an NMR-based approach, here we demonstrate that the *Xenopus laevis* Npm tail domain controls the binding of histones at its largest acidic stretch (A2) via direct competition with both the C-terminal basic stretch and basic nuclear localization signal. NMR and small-angle X-ray scattering (SAXS) structural analyses allowed us to construct models of both the tail domain and the pentameric complex. Functional analyses demonstrate that these competitive intramolecular interactions negatively regulate Npm histone chaperone activity in vitro. Together these data establish a potentially generalizable mechanism of histone chaperone regulation via dynamic and specific intramolecular shielding of histone interaction sites.

[1] Department of Biochemistry, Albert Einstein College of Medicine, 1300 Morris Park Avenue, Bronx, NY 10461, USA. [2] Department of Chemistry, Stanford University, Stanford Synchrotron Radiation Lightsource, 2575 Sand Hill Road, Menlo Park, CA 94025, USA. [3] Present address: Laboratory of Biochemistry and Molecular Biology, Rockefeller University, 1230 York Avenue, New York, NY 10065, USA. Correspondence and requests for materials should be addressed to D.S. (email: david.shechter@einstein.yu.edu)

The nucleosome is the fundamental repeating unit of chromatin, composed of 147 bp of DNA wrapped around an octamer of histone proteins, two copies each of H2A, H2B, H3, and H4[1]. Nucleosome assembly is required for compaction and organization of the eukaryotic genome. Many proteins coordinate the assembly of a single nucleosome, including ATP-dependent chromatin remodelers and ATP-independent histone chaperones[2]. Histone chaperones are a large class of proteins responsible for: (1) binding the basic histone proteins, (2) shielding histones from nonspecific interactions, (3) nuclear import of histones, and (4) facilitating the deposition of histones onto DNA either by directly transferring histones to DNA or by handing histones off to other chaperones[3–5]. Histone chaperones are structurally diverse with few commonalities in their folds[6,7]. Intrinsically disordered regions (IDRs) and acidic stretches are found in most histone chaperones and may play key roles in histone binding and charge shielding[8–10], though their roles are not well understood due to difficulties in their structural characterization.

Nucleoplasmin (Npm) is a highly conserved, embryonic histone chaperone required for the storage of H2A/H2B during vertebrate development[11–13]. Loss of Npm in female mice caused fertility defects[14], and Npm was recently identified as one of six factors capable of independently assembling mitotic chromatids[15]. Npm contains an N-terminal core pentamerization domain (residues 16–118), a C-terminal tail domain (residues 119–195), and a short N-terminal tail (residues 1–15). The crystal structure of the core domain shows that it is composed of antiparallel β-sheets and assembles into an extremely stable homopentamer[16]. The tail domain is predicted to be an IDR; however, there is very little structural information available for this region. The tail domain is necessary for Npm function and likely acts as a major histone interaction site through its largest acidic stretch (A2), which was previously hypothesized to be shielded by the C-terminal half of the tail domain[17–20]. We, and others, previously showed that Npm is heavily modified during development on both the short N-terminal tail and longer C-terminal tail, and that these modifications correlate with changes in its histone binding and deposition activities and structure[18,21]. Therefore, we hypothesized that the disordered C-terminal tail domain of Npm represents both a major histone interaction site and regulator of Npm histone chaperone function during development.

Here, we demonstrate that the *Xenopus laevis* Npm tail domain is largely disordered, yet regulates histone accessibility to its major acidic stretch (A2) via specific, electrostatic intramolecular interactions with both its basic C-terminus and basic nuclear localization signal (NLS). Using nuclear magnetic resonance (NMR), chemical shift perturbation (CSP), and paramagnetic relaxation enhancement (PRE) experiments, we show that histones mainly bind to acidic and aromatic residues in A2, disrupting regulatory intramolecular interactions causing a partial conformational opening of the tail domain. Structural modeling using NMR, molecular dynamics (MD) simulations and small-angle X-ray scattering (SAXS) data yield structural insights into the apo and histone-bound states of the tail domain and the pentameric complex. Functional analyses using Npm tail truncations reveal regions necessary for histone binding, deposition, and aggregate removal. These data highlight a dynamic mechanism in which specific electrostatic intramolecular interactions within highly charged IDRs can regulate histone chaperone function via shielding of histone interaction sites.

## Results

### The disordered Npm tail samples structured states.
The 8.6 kDa Npm tail domain contains large acidic and basic stretches and is predicted to be disordered (Fig. 1a; Supplementary Fig. 1a, b). To better study the structural states and role of the tail domain in regulating histone binding, we purified this monomeric region independent of the core pentamer (Supplementary Fig. 1c, last lane). Using $^{13}C/^{15}N$-labeled tail, we generated a $^{1}H$-$^{15}N$ heteronuclear single quantum coherence (HSQC) NMR spectrum of the domain (residues 119–195) and assigned 50 of 70 resonances visible in the spectrum by standard triple resonance approaches (Fig. 1b, c). Many of the tandem glutamate residues of A2 were unassignable due to their chemical shift degeneracy, leading to highly overlapped peaks at ~8.55 ppm in the $^{1}H$ dimension and ~122.8 ppm in the $^{15}N$ dimension. The spectrum shows low dispersion in the $^{1}H$ dimension characteristic of an IDR, with all backbone $^{1}H$ shifts lying between 7.8 and 8.8 ppm. Peaks corresponding to residues 156–162 and 179–189 are noticeably broadened (Supplementary Fig. 1d), suggestive of an intermediate-timescale conformational exchange occurring in these regions. We tracked peak intensities and $^{1}H$ line widths as a function of tail concentration, and observed linear increases in peak intensities (Supplementary Fig. 1e) and no substantial changes in line widths (Supplementary Fig. 1f), indicating that these broadened peaks likely arise from intramolecular, and not intermolecular, conformational exchange.

Using Cα, Cβ, and Hα chemical shifts, we estimated the secondary structure propensity (SSP) along the sequence of the tail domain[22]. We found that although the tail domain is largely disordered, many residues transiently sample β-sheet like conformations as evidenced by a negative SSP index (Fig. 1d). Circular dichroism of the tail domain and CONTIN secondary structure analysis[23] independently confirmed a substantial degree of β-sheet and turn contribution to the tail domain secondary structure, predicting 8.4% helical, 25.6% sheet, 22.0% turn, and 44.0% disordered composition (NRMSD = 0.080) (Fig. 1e). These data suggest that although the tail domain is largely disordered, it dynamically samples folded conformations that may be relevant to its function.

### Electrostatic interactions regulate A2 accessibility.
We previously hypothesized that the Npm tail domain may be auto-regulated through intramolecular competition between the basic C-terminus and histones for binding A2[18]. To test this model, we performed NMR CSP experiments using $^{15}N$-labeled tail with increasing concentrations of NaCl to determine if electrostatic intramolecular interactions are disrupted. Large CSPs in the fast-exchange regime were observed between residues 143–149 and residues 177–180 (Fig. 2a, b). These two regions correspond to more disordered regions in the secondary structure of the tail domain (Fig. 1d) and lie at key points between positive and negatively charged regions of the tail (charge plot, Fig. 2b). This suggests that these regions, which undergo fast-exchanging CSPs upon salt titration (centered on S144 and T179), represent "hinges" in a compact structural ensemble mediated by electrostatic intramolecular interactions.

Next, we performed analytical ultracentrifugation (AUC) sedimentation velocity studies of the tail domain as a function of NaCl concentration to determine the overall structural consequences of salt-mediated disruption of these interactions. We observed decreases in both the sedimentation coefficient (Fig. 2c; Supplementary Fig. 2a) and diffusion coefficient (Supplementary Fig. 2b) with increasing salt concentrations. Sedimentation curves fit best to a monomeric species with no concentration dependence in the sedimentation profiles, consistent with a transition from a compact to an extended conformation upon disruption of electrostatic intramolecular interactions.

To identify residues involved in interactions with histones, we performed CSP experiments using [15]N-labeled tail domain and unlabeled H2A/H2B dimers. Titration of H2A/H2B caused disappearance of peaks corresponding to residues 123–130 (Fig. 2b, d), likely indicating strong interactions between this region and the relatively large H2A/H2B dimer causing increased relaxation rates[24]. This region includes two aromatic residues (Y123 and W125) followed by four acidic residues (E127, E128, D129, and E130). Additionally, we observed CSPs in the fast-exchange regime corresponding to residues between 143 and 149 (Fig. 2a, b). These residues shifted in approximately the same direction and magnitude as in salt titration experiments, indicating that these fast-exchanging CSPs likely arise via a structural opening at the first hinge region, centered at S144, similar to NaCl titration. We observed no significant changes at the second hinge region, centered at T179, upon binding to H2A/H2B indicating that this region remains in a closed conformation

(Fig. 2a, b). Additionally, we observed nearly identical CSPs and peak disappearance when titrating histones H3/H4 (Supplementary Fig. 3a, b), indicating that both H2A/H2B and H3/H4 compete for binding A2 and consistent with prior observations of H2A/H2B-specific chaperones binding to H3/H4[25,26]. The stoichiometry of this complex was 1:1 tail:H3/H4 dimer (Supplementary Fig. 7c), and is also consistent with the recently reported binding affinity and stoichiometry of Npm to H3/H4[27].

The HSQC of the tail domain remains lowly dispersed in the [1]H dimension upon binding to H2A/H2B or H3/H4, indicating that there is no large-scale ordering of the tail domain upon histone binding. To gain insights into the dynamics of the tail domain, we measured [15]N relaxation times ($T_1$ and $T_2$) and [1]H-[15]N heteronuclear NOE ratios for residues in both the unbound and H2A/H2B-bound states (Fig. 2e). For the unbound tail (blue circles), we observe relatively large $T_1$ (500–750 ms) and $T_2$ (200–400 ms) values for most residues, as well as NOE ratios <

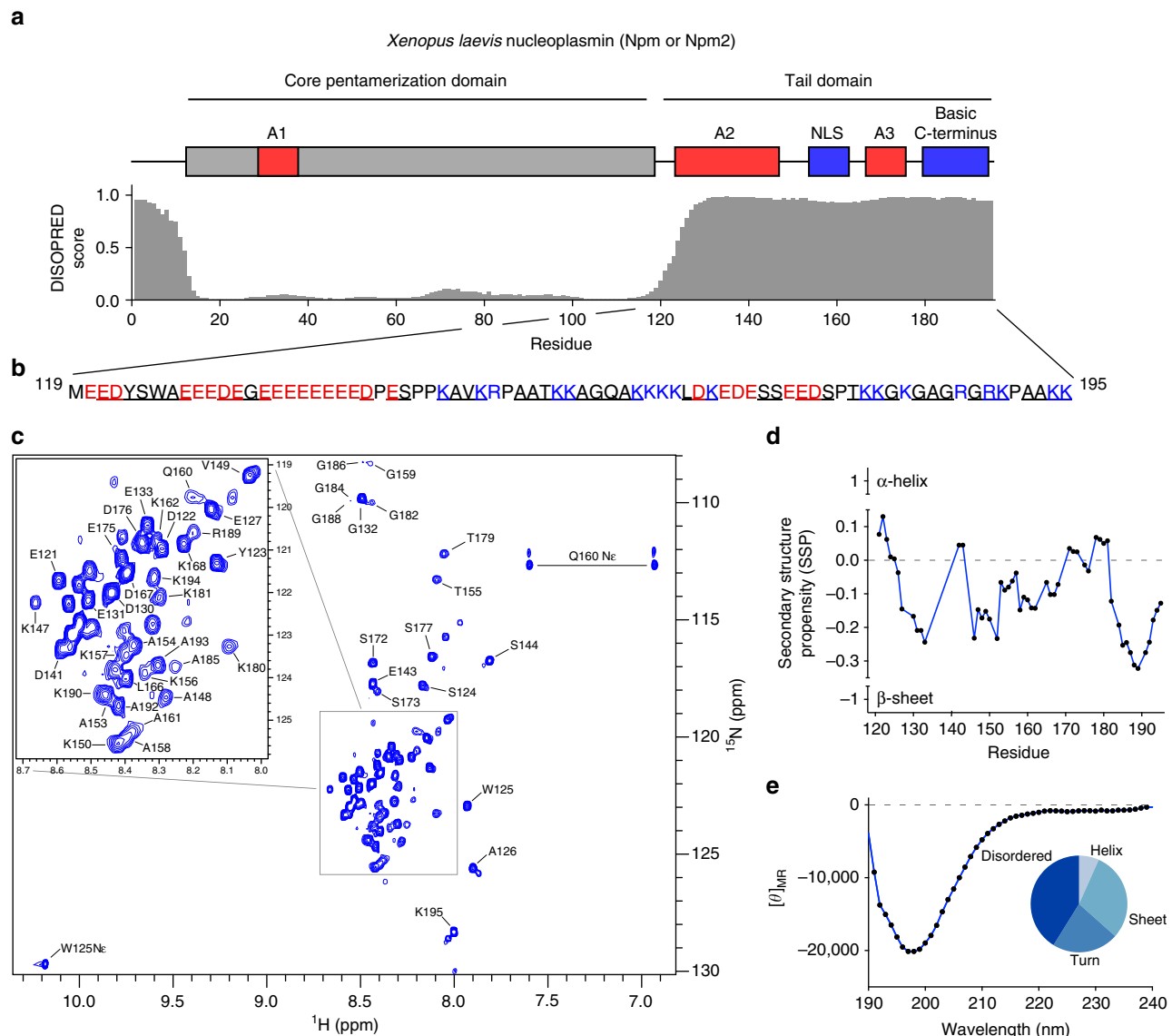

**Fig. 1** Transient structural states of the Npm tail domain detected by NMR. **a** Domain layout of *Xenopus laevis* Npm. Core domain is colored gray, acidic stretches (A1, A2, and A3) colored red, basic NLS and C-terminus colored blue. Disorder prediction of Npm by DISOPRED3 software (0 = ordered, 1 = disordered). **b** Sequence of the Npm tail domain (residues 119–195). Acidic and basic residues colored red and blue, respectively. Underlined residues are assigned in the [1]H-[15]N HSQC spectrum. **c** [1]H-[15]N heteronuclear single quantum coherence (HSQC) spectrum of the Npm tail domain. **d** Secondary structure propensity (SSP) plot of the Npm tail domain derived from Cα, Cβ, and Hα chemical shift values. +1 indicates a stable α-helix, −1 indicates a stable β-sheet. **e** Circular dichroism (CD) spectrum and CONTIN secondary structure analysis of the Npm tail domain

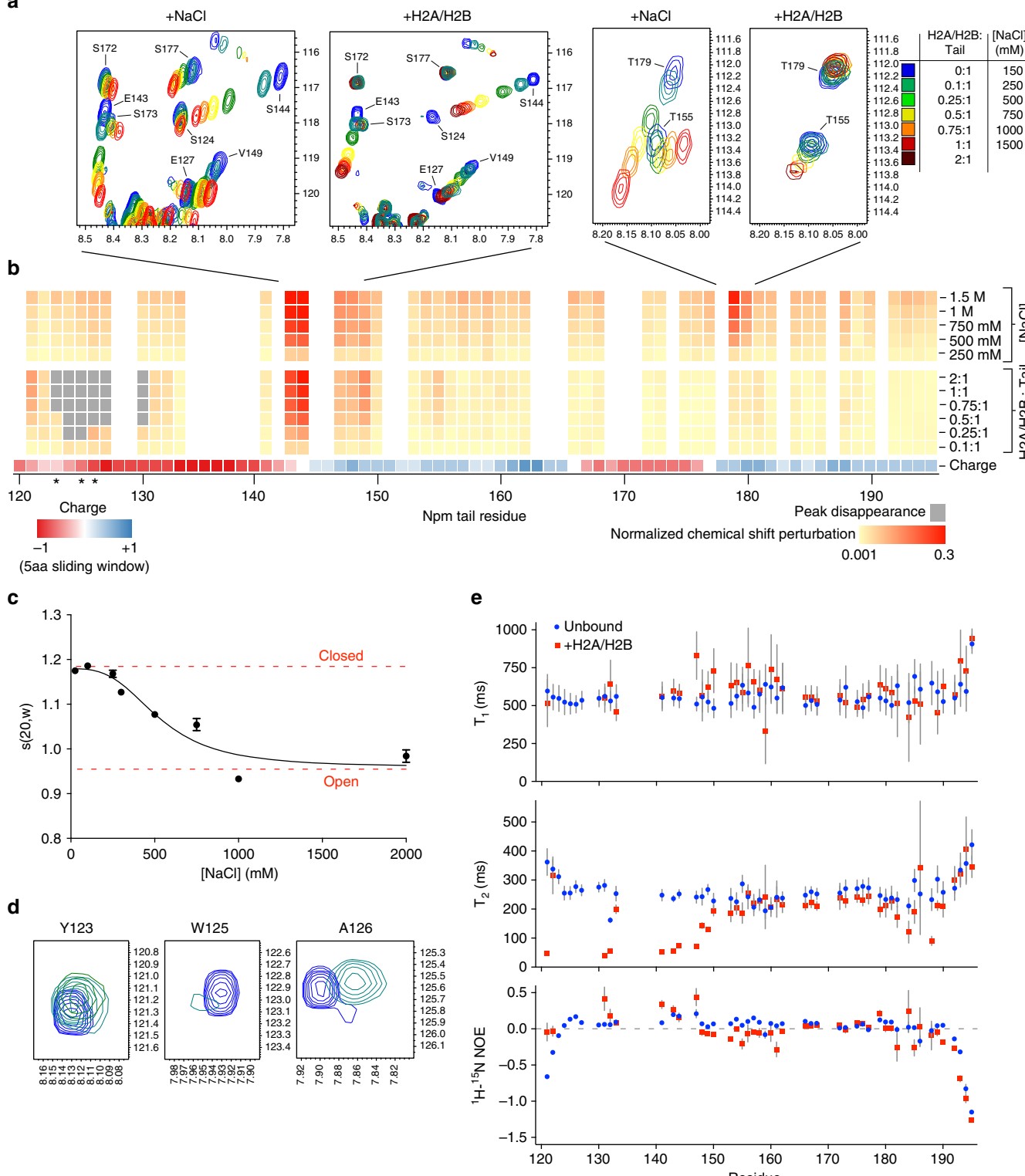

**Fig. 2** Biophysical analyses of the conformations of the Npm tail domain. **a** NMR chemical shift perturbation (CSP) example spectra upon titration of NaCl and H2A/H2B. Blue to red indicates increasing amounts of NaCl or H2A/H2B (legend shown on right). **b** Heat map of CSP upon titration of NaCl and H2A/H2B. Yellow to red in the heat map indicates increasing CSP in the spectra and gray indicates binding-induced peak disappearance (legend at bottom right). Starred residues (Y123, W125, and A126) shown in letter D. Npm tail charge displayed using a 5aa-sliding window from −1 (red) to +1 (blue). **c** Analytical ultracentrifugation (AUC) sedimentation velocity profile of the Npm tail domain (15 μM) at various concentrations of NaCl. Sedimentation values normalized to s(20,w). **d** NMR CSP example spectra of Y123, W125, and A126 upon H2A/H2B titration. Peak disappearance indicates binding to H2A/H2B. **e** Graph of [15]N relaxation times $T_1$ (top) and $T_2$ (middle), and heteronuclear [1]H-[15]N NOE ratios (bottom) of assigned residues of the tail in the unbound (blue circles) and H2A/H2B-bound (red squares) states. Standard errors (gray bars) derived from fits to single exponential decay curves for $T_1$ and $T_2$ values, and from propagated signal-to-noise for [1]H-[15]N NOE values

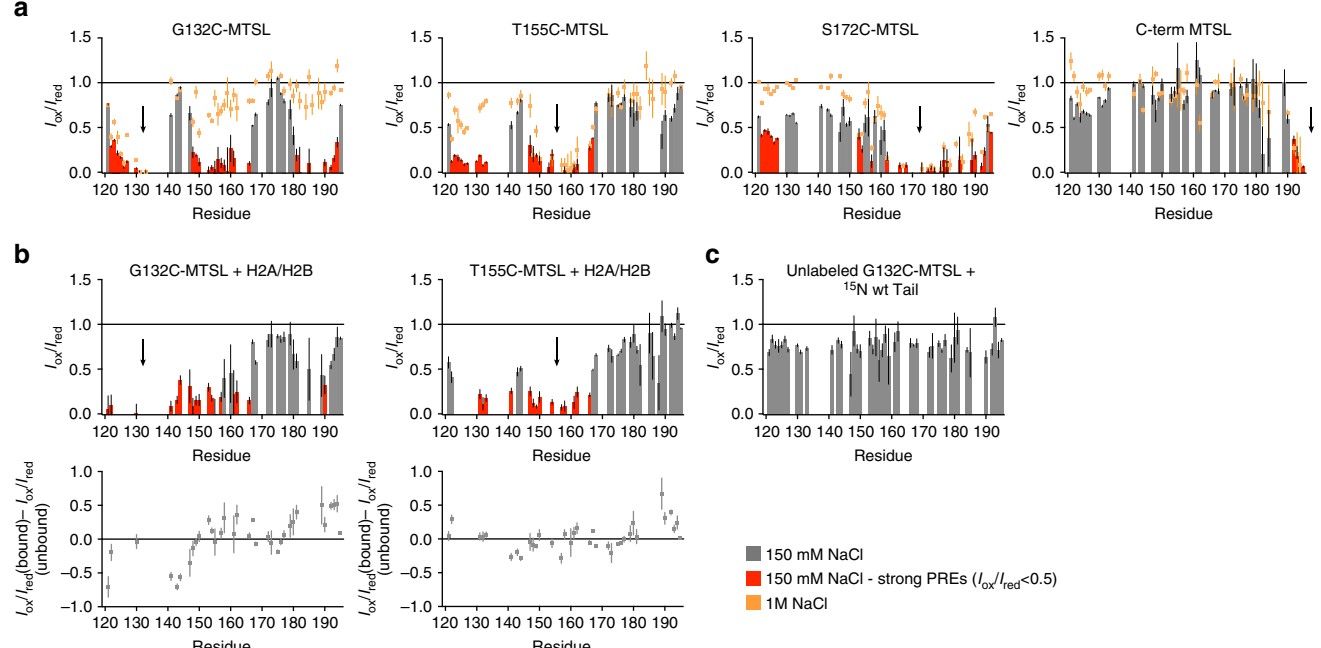

**Fig. 3** Paramagnetic relaxation enhancement NMR (PRE-NMR) analysis of intramolecular contacts. **a** PRE intensity ratio graphs ($I_{ox}/I_{red}$) of the Npm tail domain with MTSL paramagnetic spin labels at four different positions (positions indicated by arrows) and at two different NaCl concentrations (gray/red bars = 150 mM NaCl, orange points = 1 M NaCl). Error bars are inversely proportional to the propagated signal-to-noise ratio of individual resonances. Significant PRE effects ($I_{ox}/I_{red} < 0.5$) in the low-salt condition are highlighted in red. **b** Upper: $I_{ox}/I_{red}$ graphs of the Npm tail domain bound to H2A/H2B. Two sets of intramolecular PRE effects derived from the complex (positions of spin labels indicated by arrows). Error bars are inversely proportional to the propagated signal-to-noise ratio of individual resonances. Same coloring scheme used as in **a**. Lower: Difference maps in $I_{ox}/I_{red}$ values from the unbound and histone-bound states. Positive values indicate that residues are further from the spin label position after histone binding, negative values indicated that residues are closer to the spin label position after histone binding. **c** Intermolecular PRE effects measured using 1:1 molar ratio of non-isotope-labeled G132C-MTSL tail: $^{15}$N-labeled tail at 200 μM total protein concentration. Lack of strong PRE effects indicate very few or only transient intermolecular interactions. Error bars are inversely proportional to the propagated signal-to-noise ratio of individual resonances

0.25 for all residues, consistent with the tail being a disordered protein. Upon binding to H2A/H2B (red squares, 1:1 molar ratio), the largest changes were observed in the decreased $T_2$ values of assigned N-terminal residues 121–149. Slight increases in NOE ratios at the N-terminus were also observed upon binding to H2A/H2B, though very few changes in $T_1$ values were observed for these residues. These data indicate that H2A/H2B binding leads to slower dynamics for backbone NH groups of assigned residues between 121 and 149, though not a complete ordering of this region. This is likely caused by increased rotational correlation time of A2 upon binding to the relatively large H2A/H2B dimer (Supplementary Fig. 2c). Residues 150–195 appear approximately equally dynamic in both the unbound and H2A/H2B-bound states. Due to the peak disappearance upon binding and unassignable residues, it is possible that small regions in A2 may undergo binding-induced folding. Taken together with CSP experiments, these data indicate that the Npm tail domain dynamically samples electrostatic intramolecular interactions, undergoes a partial opening upon binding to histones, and that histone binding leads to a loss of conformational freedom for residues surrounding A2.

**Paramagnetic relaxation enhancement and NMR structural analysis.** We next structurally characterized all long-range intramolecular interactions regulating A2 accessibility and histone binding. We made four cysteine mutants along the Npm tail domain: G132C, T155C, S172C, and a C-terminal cysteine addition (C196), and conjugated S-(1-oxyl-2,2,5,5-tetramethyl-2,5-dihydro-1H-pyrrol-3-yl)methyl methanesulfonothioate (MTSL)

paramagnetic spin labels at each position (Supplementary Fig. 4a). In these experiments, spin labels cause increased relaxation rates for amide protons on residues <~25 Å away, leading to peak broadening and reduction of intensity in the HSQC spectrum (Supplementary Fig. 4b)[28].

At 150 mM NaCl, we observed strong PREs from G132C-MTSL to two regions in the tail domain: the nuclear localization signal (NLS) (residues 150–165) and basic C-terminus (residues 180–194) (Fig. 3a first panel, Supplementary Fig. 4c). We observed reciprocal strong PREs from T155C-MTSL toward A2 and weaker PREs toward the basic C-terminus, indicating that these three regions are close in space. We also observe broad PREs across most residues of the tail domain from S172C-MTSL, including strong effects toward residues 122–127. Weak PREs were observed on residues 123–131 from the C-terminal MTSL, which is not surprising given the terminus' fast dynamics (Fig. 2e). We performed the same PRE experiments at 1 M NaCl and observe a loss in long-range PRE effects for all spin labels, consistent with a conformational opening of the tail domain at 1 M NaCl due to disruption of electrostatic intramolecular interactions (Fig. 3a, orange squares).

We also used the $^{15}$N-labeled tail tagged at G132C-MTSL and T155C-MTSL to directly measure intramolecular conformational changes upon binding to H2A/H2B dimers. For both spin label positions, we observed a decrease in intramolecular PREs for the basic C-terminus (residues 184–193) when bound to H2A/H2B (Fig. 3b, difference plot shown below) compared to the unbound protein (Fig. 3a), providing direct structural evidence of a conformational opening upon binding. PREs from G132C-MTSL to the NLS and reciprocal PREs from T155C-MTSL to

A2 were not significantly different from the unbound tail, indicating that these regions do not fully dissociate upon binding H2A/H2B, consistent with the partial unfolding observed in NMR CSP experiments (Fig. 2b). Residues 121–122 and 141–144 experience stronger intramolecular PREs from G132C-MTSL upon binding to histones (Fig. 3b, difference map), indicating that these residues are either closer to the spin label position or exhibit decreased dynamics upon binding. Finally, to determine whether these effects were due to intra or intermolecular interactions, we mixed $^{15}$N-labeled tail domain (not MTSL tagged) with unlabeled tail tagged at G132C at 1:1 stoichiometry in 150 mM NaCl (Fig. 3c). We did not observe strong PREs, as in the first panel of Fig. 3a, indicating that the effects observed in Fig. 3a are mainly due to intramolecular, not intermolecular, interactions. However, transient intermolecular interactions are likely, given that many residue $I_{ox}/I_{red}$ values are <1.

**Ensemble structural modeling of the Npm tail domain.** To model the structural states sampled by the tail domain, we performed ensemble calculations using a simulated annealing protocol in Xplor-NIH software in which the PREs are averaged over a given ensemble size. We first determined the minimum number of conformers necessary to fit the experimental PRE data (Fig. 3a, low salt) to avoid over-fitting the data. We calculated 50 ensembles with a variable number of conformers per ensemble (350 ensembles, 1600 conformers calculated total) and assessed the quality of the fits from the top five ensembles in each group (Fig. 4a). The PRE Q-factor, a measure of how well the back-calculated PREs from the ensemble match the observed PREs, decreased significantly upon increasing the ensemble size from a single conformer to two conformers, indicating a better fit. Further increasing the ensemble size from 2 to 10 conformers yielded only marginally better fits to the PRE data, and, as expected, results in many more extended conformers in the ensemble that contribute little to the observed PREs. This indicates that a two-conformer ensemble is sufficient to model the observed PREs without over-fitting the data.

Analysis of the top five two-conformer ensembles (Fig. 4b) shows that all conformers adopt a folded back state with varying

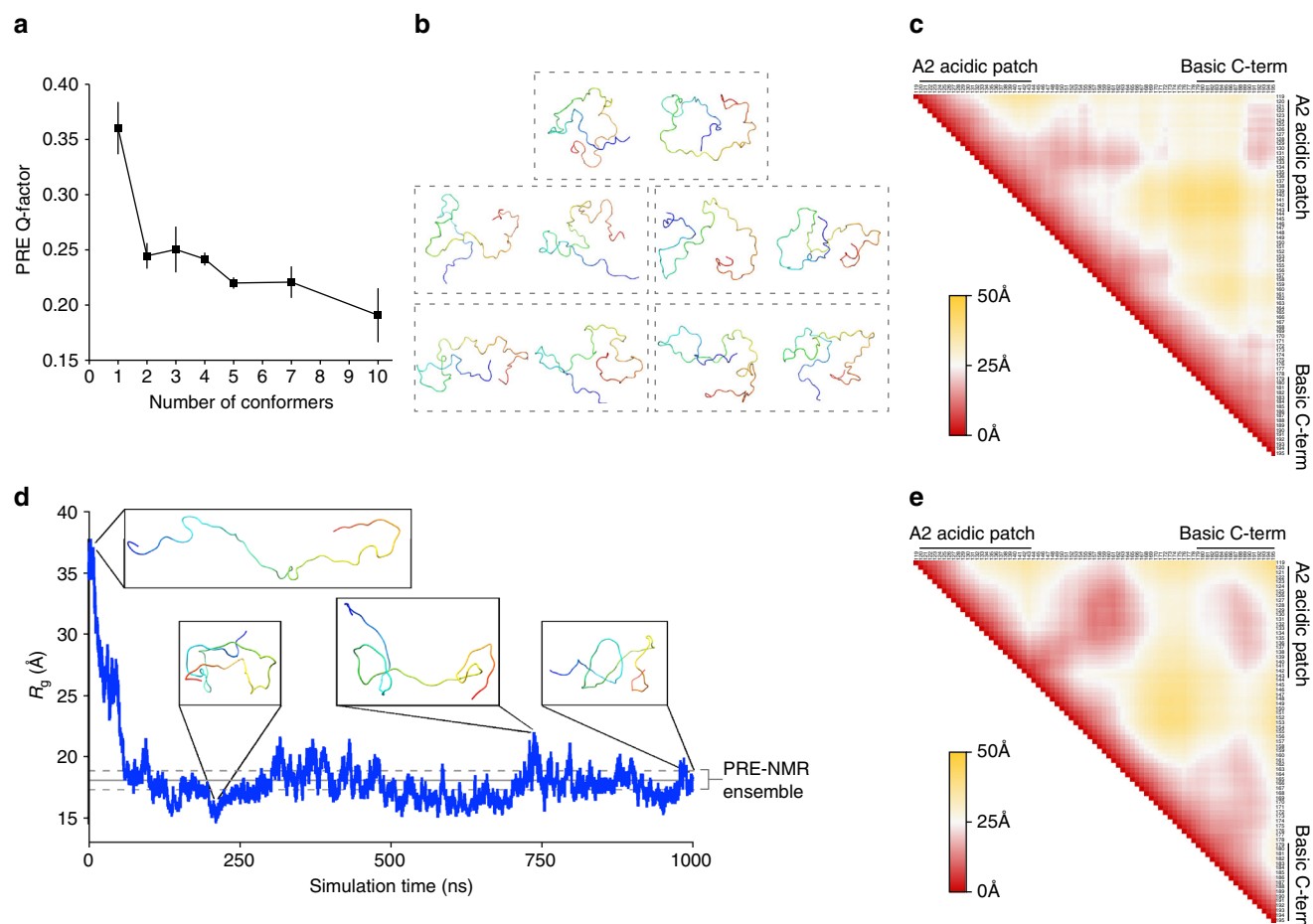

**Fig. 4** Ensemble structural modeling and MD simulation of the Npm tail domain. **a** PRE Q-factor values with standard errors from the top five ensembles out of 50 with a variable number of conformers per ensemble ranging from 1 to 10. 350 ensembles corresponding to 1600 conformers were calculated in total. **b** Top five PRE-derived two-conformer ensemble models of the Npm tail domain. Each two-conformer ensemble is boxed. Conformers colored blue to red from N-terminus to C-terminus. **c** Protein Cα–Cα contact map of the tail derived from the top five two-conformer PRE models in **b**. Axes are residues of the tail domain from N to C-terminus from left to right and top to bottom. Colored red to yellow by average distance ranging from 0 to 50 Å, respectively. **d** 1 μs unrestrained all-atom molecular dynamics (MD) simulation of the Npm tail domain starting from an extended conformation. Radius of gyration ($R_g$) measured as a function of simulation time. Representative structures show the starting, the most compact, and the most extended, and the final conformations observed during the simulation. Gray lines are the average and standard deviation of the $R_g$ measured from the PRE-NMR 2-conformer ensemble (**b**). **e** Time-averaged protein Cα–Cα contact map from the 1 μs MD simulation using an extended starting model. Same axes and coloring scale as in **c**

degrees of compactness, with the average radius of gyration ($R_g$) of the ensemble = $18.07 \pm 0.78$ Å. We generated an average Cα–Cα contact map from these two-conformer ensembles to better understand their intramolecular contacts (Fig. 4c). Though many intramolecular interactions in individual conformers average out in the contact map derived from the diverse ensemble, there are clear intramolecular contacts between residues 121–134 in A2 and residues 189–195 of the basic C-terminus. We also observe a "hinge" formed by contacts made between residues 128–136 in A2 and residues 149–165 in the basic NLS. Contacts between the NLS (residues 147–157) and the basic C-terminus (residues 189–193) are evident although less pronounced, suggesting that these regions are in proximity to one another in the ensemble due to the direct contacts formed by A2 to NLS and A2 to basic C-terminus.

To determine if the intramolecular interactions observed in the ensemble are stable and to gain further insights into the dynamics of the tail domain, we performed a restraint-free 1 μs MD simulation of the tail domain using an extended conformation as a starting structure ($R_g = 34.65$ Å) and the TIP4P-D water model optimized for the simulation of disordered proteins[29]. The MD trajectory shows a rapid collapse in the structure within the first 100 ns (Fig. 4d), leading to a time averaged $R_g = 18.16 \pm 2.85$ Å, very similar to the value observed for the two-conformer NMR ensemble (Fig. 4d, shown as gray line). The time-averaged Cα–Cα contact map from this simulation (Fig. 4e) shows extremely similar intramolecular contacts compared to the two-conformer PRE ensemble (Fig. 4c), indicating agreement between the two independent methods of modeling the structural ensemble of the tail domain. The similarities in $R_g$ values and intramolecular contacts formed in the PRE-NMR ensemble and an unrestrained MD simulation suggest that A2 to NLS and A2 to basic C-terminus autoregulatory intramolecular contacts are highly dynamic between multiple residues in each region, but together lead to a stable compaction of the Npm tail in the unbound state.

**Mechanisms of H2A/H2B binding by the Npm tail domain.** To identify the H2A/H2B region that the tail domain recognizes, we employed a comprehensive histone peptide array (JPT Peptide Technologies). Full-length Npm (NpmFL, 1–195), the core domain (1–118), or a GST-tagged tail domain (119–195) were incubated with 3868 unique histone peptides on a chip and binding was observed by immunofluorescence. NpmFL and the GST-tail showed a very similar binding pattern corresponding to H2A/H2B binding mainly through residues 16–40 of H2A, with more minor contribution by residues 80–100 of H2B (Fig. 5a). We also observed interactions with H3 and H4, particularly on the C-terminus of H3 (residues 115–130) and the N-terminus of H4 (residues 20–50) (Supplementary Fig. 5a), consistent with the interactions observed between A2 and H3/H4 in NMR CSP experiments (Supplementary Fig. 3a, b). The region of H2A that the tail domain recognizes (residues 16–40) makes direct contacts with the phosphate backbone of DNA in the nucleosome structure (Supplementary Fig. 5b), suggesting that A2 may compete with DNA for histone interaction. We saw few interactions from the core domain in this assay, indicating that A2 of the tail domain represents a major histone binding site on Npm, consistent with the previously reported weak affinity of the core domain toward histones[19].

We next used PRE-NMR to determine the site on the H2A/H2B dimer bound by A2 in solution. H2A/H2B was labeled with the MTSL paramagnetic spin label at four introduced cysteines around the structure of the dimer: H2A E64C, H2B S61C, H2B G72C, and H2B T116C (Fig. 5b). We observed PREs to residues in the [15]N-labeled tail domain upon binding to various MTSL-labeled H2A/H2B dimers at 400 μM (Fig. 5c) and 200 μM (Supplementary Fig. 5c, d) concentrations. Significant PREs were observed from H2A E64C-MTSL to residues 149–153 and residues 180–194 of the tail domain, indicating that the NLS and basic C-terminus are near this position in the bound state. H2B S61C-MTSL showed strong PREs to residues 131 and 132 of the tail domain. Effects from H2B G72C-MTSL were less pronounced, and effects from H2B T116C-MTSL were mainly to residues 140–150 of the tail domain.

Since the A2 region of the tail domain becomes significantly more ordered upon binding to H2A/H2B (Fig. 2e), we used these PRE effects to compute average distances from each spin label on the H2A/H2B dimer to amide protons in the A2 region of the tail domain (residues 119–149) and calculated 1000 structures of the A2:H2A/H2B complex using a simulated annealing protocol in Xplor-NIH software. We aligned the top scoring 100 structures to account for the diversity in the ensemble of structures, and the positions of A2 on the H2A/H2B dimer were mapped as a reweighted atomic probability density map (Fig. 5d). In this representation, 50% of A2 conformers lie within the blue region, and 90% lie within the red region around the H2A/H2B dimer. Although an exact atomic position for A2 cannot be determined from these data, the majority of the A2 conformers (81%) are positioned directly adjacent to αN and α1 helices of H2A, consistent with the contacts observed in the peptide array (Fig. 5a). We also observed a smaller subset of conformers (19%) bound on the opposite side of the H2A/H2B dimer, adjacent to α3 of H2B, that are also consistent with the observed PREs. Together, these results suggest that A2 in the Npm tail domain binds to H2A/H2B dimers by making contacts with the N-terminal region of the histone fold of H2A, but may sample other binding sites such as the C-terminal region of the histone fold of H2B.

**Structural analysis of the full-length pentameric protein.** We next tested if the Npm tail domain adopts similar conformations in the full-length protein (NpmFL). NpmFL has five tails directly adjacent to one another at the pentameric distal face (schematic in Supplementary Fig. 1b), which could facilitate alternative intra or intermolecular interactions. A [1]H-[15]N HSQC of NpmFL at 150 mM NaCl shows very similar peak positions to most residues in the tail domain alone at the same salt concentration (Fig. 6a). Residues from the core domain of NpmFL are not visible in the spectrum likely due to the slow tumbling of this ~65 kDa pentameric domain relative to the fast motions of the disordered tails. Regions that shift significantly compared to the monomeric tail include residues at the N-terminus of the tail domain whose chemical shift is likely altered by the adjacent core domain (W125 and A126) and from both hinge regions (E143, S144, K147, A148, V149, T179, K180, and K181). Both hinge regions shift in the same direction as with NaCl titration of the monomeric tail (Fig. 2b), and appear closer in position to the monomeric tail at 250 mM NaCl (Fig. 2a), suggesting that the tails of NpmFL adopt a similar, but slightly more open conformation compared to the isolated tail domain. [1]H line widths for well-separated peaks of the monomeric tail and NpmFL showed that while residues at the base of the tails in NpmFL (E121, D122, Y123, S124, W125, and A126) are significantly broadened due to the slow tumbling of the adjacent core domain, more C-terminal residues appear as sharp as the tail domain alone (Fig. 6b). We conclude that the tails on the NpmFL pentamer do not significantly interact with the core domain, and that adjacent tails do not interact more than the transient interactions observed between tails in solution (Fig. 3c).

To gain structural insights into the complex of the Npm pentamer bound to five H2A/H2B dimers, we employed SAXS. We purified the complex of core+A2 truncation (1–145) bound

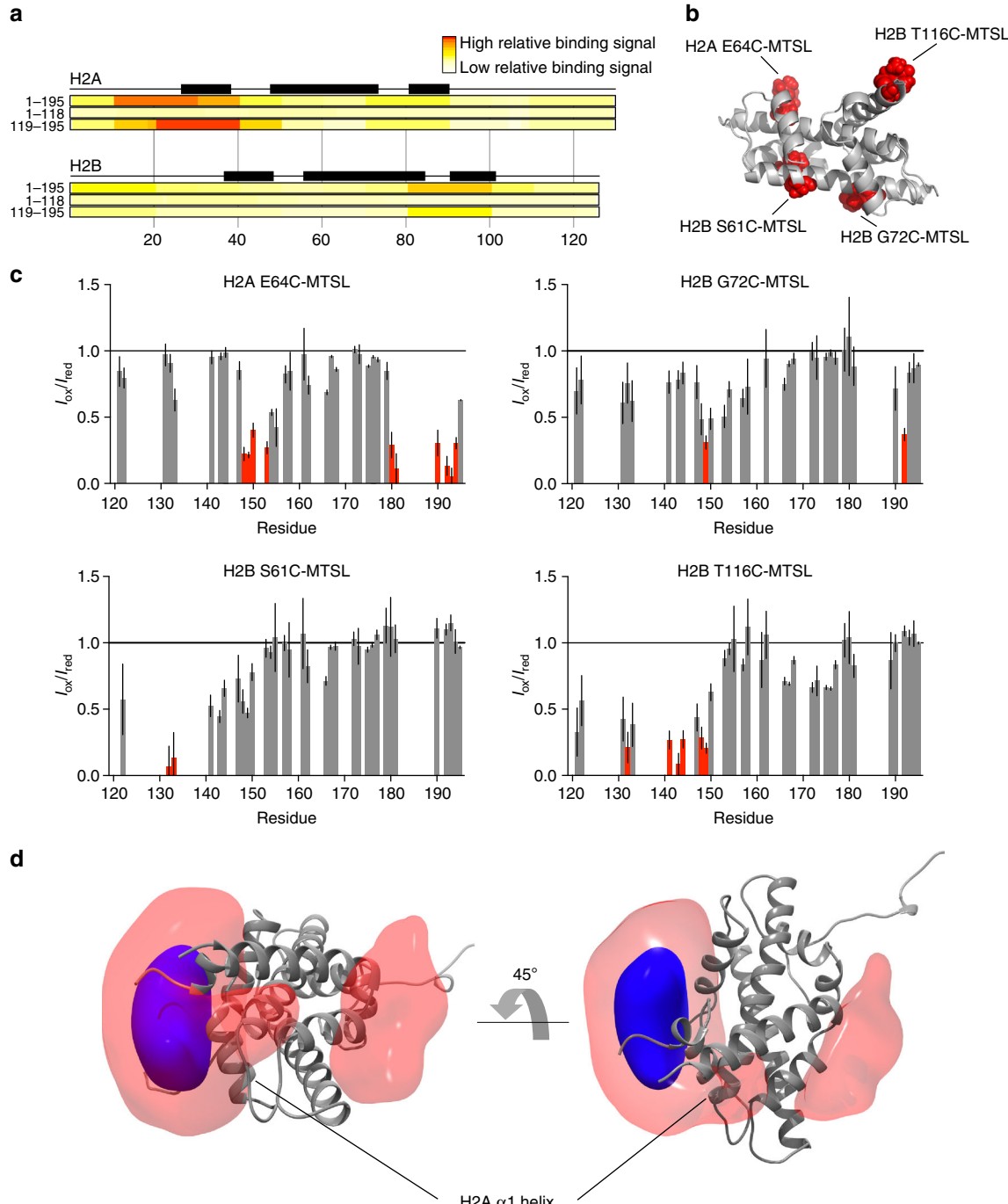

**Fig. 5** Mechanisms of H2A/H2B binding by the Npm tail domain. **a** Histone peptide array result using three different Npm truncations probed against comprehensive H2A/H2B peptides on a chip. Top = full-length Npm (1–195), middle = Npm core (1–118), bottom = GST-tagged Npm tail (119–195). Black boxes represent positions of α-helices in the histone fold. **b** Positions of MTSL paramagnetic spin labels on the H2A/H2B dimer structure. Histone tails removed for clarity. **c** $I_{ox}/I_{red}$ graphs of the Npm tail domain bound to H2A/H2B at 400 μM concentration. Four sets of intermolecular PRE effects derived from the complex. Error bars are inversely proportional to the propagated signal-to-noise ratio of individual resonances. Same coloring scheme as used in Fig. 3a. **d** Reweighted atomic probability density maps indicating the most likely positions of A2 around the H2A/H2B dimer (gray ribbon, histone tails removed for clarity). Transparent red contoured at 10%, and solid blue contoured at 50%. Two distinct binding sites are consistent with the PRE data

to five H2A/H2B dimers by size-exclusion chromatography, reasoning that this complex would be more stable than the previously analyzed NpmFL:H2A/H2B complex[19] due to the increased affinity of H2A/H2B for A2 compared to the full tail, and have decreased flexibility due to truncation of the flexible basic C-terminus (Fig. 2e). The scattering curve of the complex is shown in Fig. 6c (purple circles). Guinier and Kratky plots

indicate that the complex is stable, globular and without significant flexibility (Supplementary Fig. 6c, d), and Porod volume estimation is consistent with an Npm pentamer bound to five H2A/H2B dimers (Supplementary Table 1). Ab initio SAXS envelopes built assuming five-fold symmetry of the particles show an oblate star-shaped structure (Fig. 6d, envelope), similar to previous models[19].

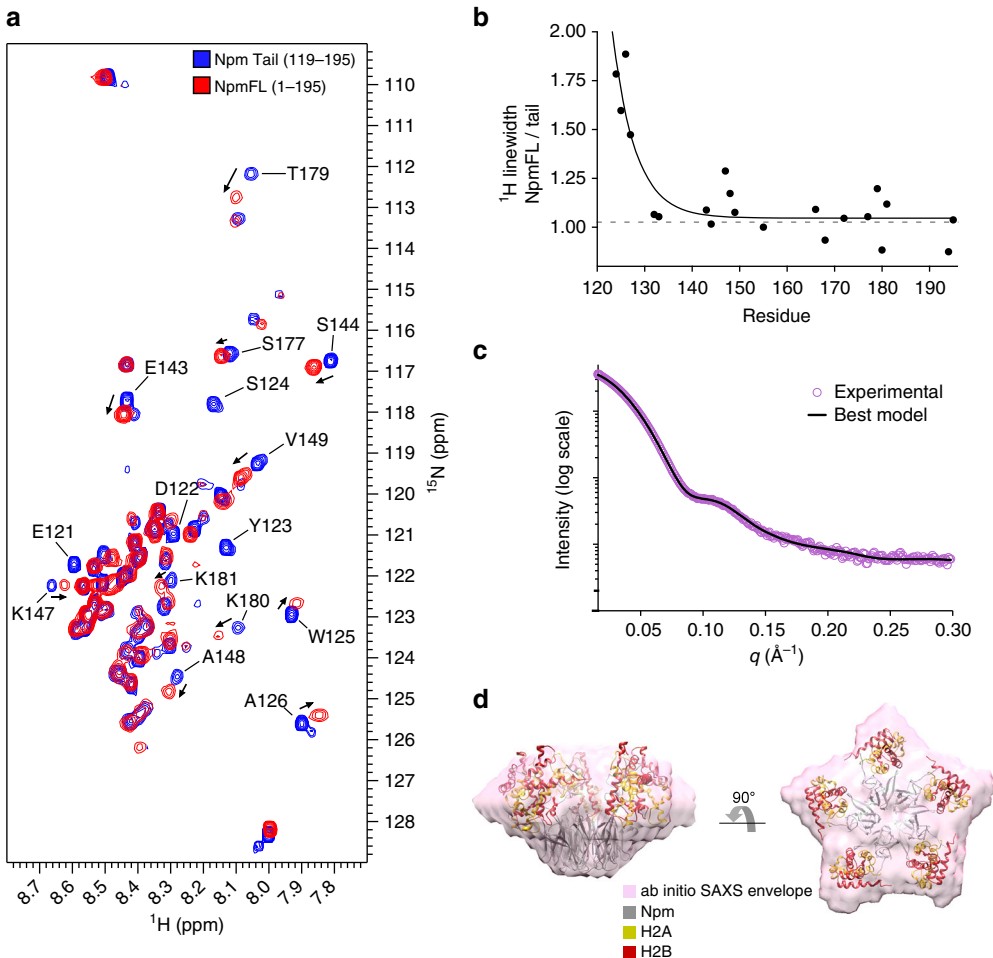

**Fig. 6** Structural analysis of the pentameric Npm unbound and bound to H2A/H2B. **a** $^1H$-$^{15}N$ HSQC of the full-length Npm (red) overlaid with the tail alone (blue) under the same buffer and temperature conditions. Residues with significant differences in chemical shift values or those broadened in the full-length protein are labeled. **b** Ratio of $^1H$ line widths of well-separated residues of the full-length protein compared to the tail domain alone. Ratio of 1 indicates peaks that are not broadened in the full-length spectrum. **c** SAXS curve (purple circles) of the core+A2:H2A/H2B complex. Top NMR-restrained SAXS hybrid model fit to the scattering curve (black line). **d** SAXS ab initio envelope of the core+A2:H2A/H2B complex (purple). Top NMR-restrained SAXS hybrid model fit into the envelope. Npm, H2A, and H2B colored gray, yellow, and red, respectively

We next built a model of the pentameric complex that is consistent with both NMR and SAXS data. We fixed A2 of the tail domain with respect to H2A/H2B within the highest density of our PRE-derived probability density map (Fig. 5d). This region is in similar positions in most of our models, and is supported by the histone peptide array (Fig. 5a), providing confidence in its approximate location relative to the H2A/H2B dimer. We included the core domain structure (PDB: 1K5J) and allowed the first 5 aa of A2 to remain as a flexible linker between the C-terminus of the core domain and the remainder of A2, while allowing the A2:H2A/H2B complex to move independently of the core during the model building. This structural modeling assumes that the structures of the core domain and histone dimer do not significantly change upon histone binding, and that the position of A2 is similar in both the tail:H2A/H2B complex and the core+A2:H2A/H2B complex.

Final NMR-restrained SAXS hybrid models fit well to the scattering curve (Fig. 6c; Supplementary Fig. 6g, i) and envelope of the core+A2:H2A/H2B complex (Fig. 6d; Supplementary Fig. 6f, h). This model positions H2A/H2B on the interface between lateral and distal faces of the pentamer, with no clashes between adjacent H2A/H2B dimers (Fig. 6d, best model fit into envelope). This model also predicts the involvement of the first

acidic stretch (A1) in the core domain in histone binding, supporting previous reports showing that mutation of A1 leads to a reduction in Npm-mediated sperm chromatin remodeling[30]. Finally, this positioning also provides a large degree of charge complementarity between Npm and H2A/H2B. Together, these data suggest that the NMR structural analysis of the monomeric tail domain in the unbound and H2A/H2B-bound states are highly relevant to the pentameric protein and complex.

**Functional analysis of Npm truncations**. To functionally test the model of Npm tail intramolecular regulation, we performed in vitro histone binding, deposition, and aggregate removal assays using a variety of Npm tail truncations (Fig. 7a; Supplementary Fig. 1c).

To quantitatively measure the interaction between the tail domain and histones, we exploited changes in W125 fluorescence upon binding to histones (Fig. 7b; Supplementary Fig. 7a, b). We quantified dissociation constants ($K_D$) and observed ~6.5× higher affinity for H2A/H2B over H3/H4. The A2 peptide (residues 119–145) binds H2A/H2B too tightly to precisely quantify by this assay, but certainly <50 nM (Fig. 7c), indicating that the C-terminal portion of the tail domain negatively

regulates histone binding. When we tested the peptide corresponding to residues whose peaks disappeared in NMR CSP experiments (residues 122–131), we observed lower affinity toward H2A/H2B. This peptide retained the aromatic residues, but has fewer acidic residues compared to both the tail domain and A2. Taken together with NMR–CSP and relaxation experiments (Fig. 2a, b, e), these results indicate that though this region makes direct contacts with H2A/H2B, the remainder of A2 is necessary for the high affinity of the tail domain toward H2A/H2B.

To test the relative affinities of pentameric tail domain truncations toward H2A/H2B dimers, we used a competitive pull-down assay with NpmFL as well as core+A2 (1–145) truncations and additional truncations of the basic C-terminus

(ΔCterm10 residues 1–185 and ΔCterm16 residues 1–179). Immobilized StrepII-tagged H2A/H2B dimers enriched for all Npm truncations containing A2 (Fig. 7d, lanes 9–12), consistent with previous pull-down experiments[18]. We tested competition in binding by mixing each truncation at a 1:1 molar ratio along with StrepII-H2A/H2B (Fig. 7d, lanes 13–18). From these experiments, we inferred the order of affinities as core+A2 (1–145) > ΔCterm10 (1–185)~ΔCterm16 (1–179) > NpmFL (1–195). This result is consistent with both A2 to NLS, and A2 to basic C-terminus intramolecular interactions negatively regulating histone binding.

Chromatin assembly assays showed that the tail domain alone has no histone deposition activity (Fig. 7e) nor did the core

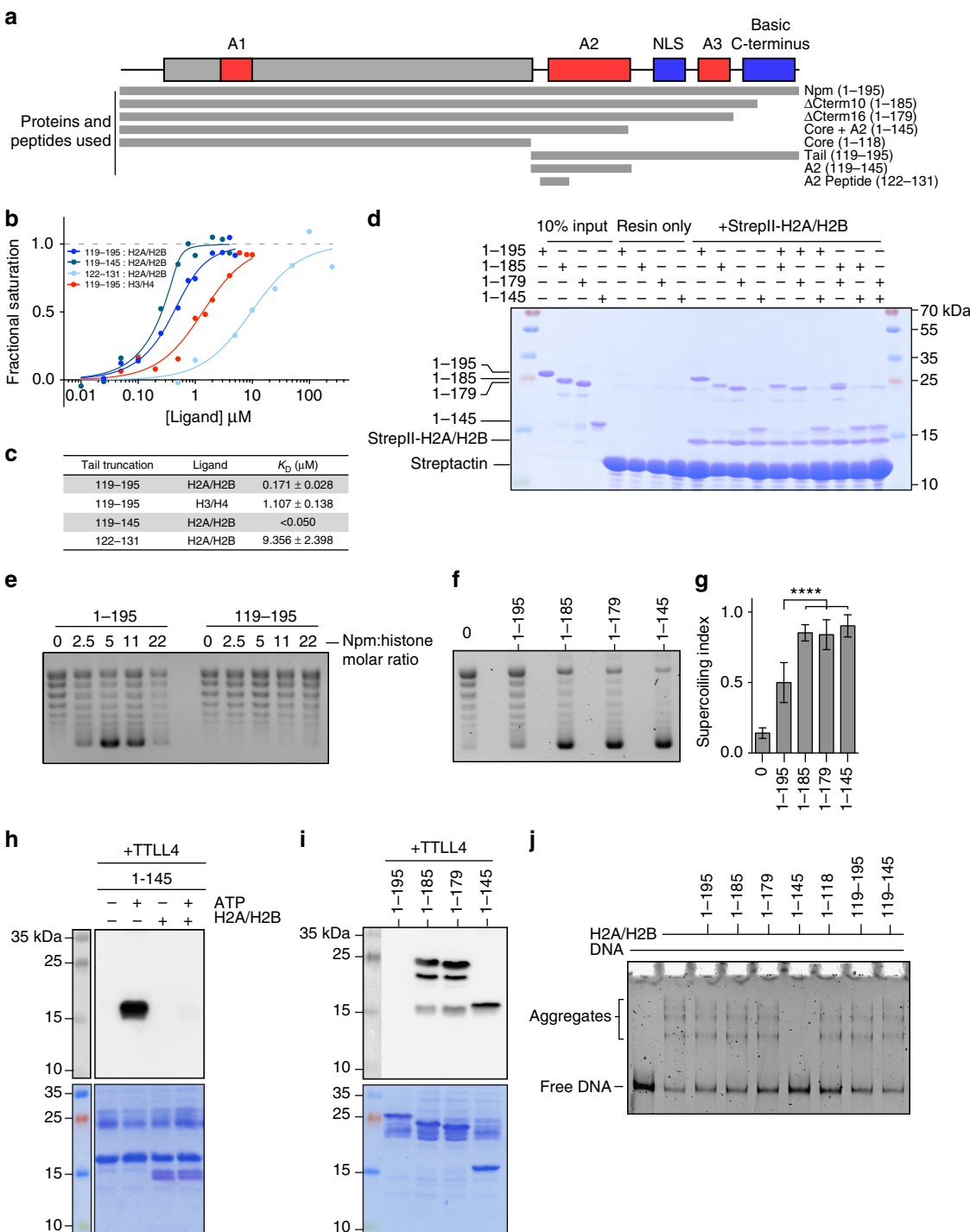

domain in previous studies[18], indicating that the tail domain is necessary but not sufficient for the chromatin assembly activity of Npm. Chromatin assembly assays using ΔCterm10 (residues 1–185) and ΔCterm16 (residues 1–179) truncations showed significantly increased histone deposition compared to NpmFL (Fig. 7f, g), consistent with a negative regulatory function for the C-terminal 10 amino acids.

We next tested these C-terminal truncations for their ability to be glutamylated by the catalytic domain of the enzyme tubulin tyrosine ligase-like 4 (TTLL4Δ526), as this assay is an effective readout of A2 solvent accessibility. We previously showed that the core+A2 truncation, but not the NpmFL, is a good substrate for TTLL4 on E127, E128, E129, and E131[18]. Addition of H2A/H2B dimers resulted in loss of glutamylation of the core+A2 truncation due to binding-induced shielding of these residues (Fig. 7h), consistent with peak disappearance observed in NMR–CSP experiments (Fig. 2b, d). Both ΔCterm10 (1–185) and ΔCterm16 (1–179) are good substrates for TTLL4, indicating that the C-terminal 10 amino acids are directly involved in limiting accessibility to A2 (Fig. 7i).

To test the role of the tail domain in directly competing for H2A/H2B:DNA interactions, we performed H2A/H2B aggregate removal assays using various truncations of the Npm tail domain. We formed aggregates of a ~500 bp linear DNA with H2A/H2B dimers at 150 mM NaCl and added a 2:1 molar excess of Npm: H2A/H2B[31,32]. In these assays, the core+A2 truncation (1–145), but not any of the other truncations, completely removed aggregates from DNA (Fig. 7j, lane 6). The tail domain and the A2 peptide alone did not have this activity (Fig. 7j, lanes 8–9) indicating that A2 is necessary, but not sufficient for the competition between of Npm and DNA for H2A/H2B binding. Together these results and chromatin assembly assays show that truncation of the basic C-terminus leads to a partial rescue of Npm histone deposition and aggregate removal activities, consistent with multiple competitive intramolecular interactions between A2 to NLS and A2 to basic C-terminus inhibiting histone binding and chaperone function. In addition, these results demonstrate that although A2 is a major site of histone interaction, the core domain is required for the histone deposition and aggregate removal activity of Npm.

## Discussion

IDRs are necessary for both the function and regulation of many cellular proteins[33]. A previous proteome-wide functional analysis showed that nuclear and chromatin binding proteins are predicted to be enriched in IDRs, suggesting that IDRs play central roles in regulating nuclear processes including chromatin structure[34]. Histone chaperones are enriched in both disordered regions and acidic stretches[10].

Here we showed that dynamic long-range structure within the Npm tail functions to regulate histone binding at its largest acidic stretch via specific, electrostatic intramolecular interactions. The C-terminal basic region, which makes key contacts with A2 that serve to inhibit histone binding, shares homology to the N-terminal tail of H2A, specifically in the glycine, arginine, and lysine repeats, suggesting that it may act as a pseudo-ligand. Additionally, family member nucleophosmin (Npm1 or B23) autoregulates its RNA-binding activity by similar intra and intermolecular interactions[35].

IDRs are easily perturbed by post-translational modifications (PTMs) due to the low-energy barriers between multiple structural states[36], allowing dynamic structural regulation. We, and others, showed that the Npm tail domain is heavily modified during embryogenesis by multiple phosphorylations, arginine methylation, and glutamylation[18,37] and that PTMs, including S144 phosphorylation, altered its conformation and function. Here, we showed that S144 lies on a hinge region that opens to allow binding of H2A/H2B, consistent with a dynamic PTM switch in adjusting tail structure to a more open state to alter histone interactions in vivo.

Glutamylation, the isopeptide addition of a glutamate onto the side chain of a primary glutamate, occurs on Npm E127, 128, 129, and 131[18]. Here, we showed that these four glutamates directly interact with H2A/H2B, making it likely that glutamylation increases histone binding affinity and/or specificity by providing both extra negative charge and bulk to this region. In addition, the branched structure of glutamylation may act to alter the local conformation and/or dynamics of the tail domain to enhance histone binding. Glutamylation occurs on acidic stretches of many other histone chaperones and acidic nuclear proteins, suggesting a broader role in regulating histone interactions[38,39].

We show here that residues [123]YSWAEEED[130] disappeared in the HSQC upon binding to histones, indicating a strong and direct interaction. This region is composed of both acidic and aromatic amino acids, is highly conserved within vertebrates, and is remarkably similar to H2A/H2B-binding sequences recently identified in multiple other histone chaperones including: FACT, SWR1, ANP32E, Nap1, and Chz1[40–44]. All of these chaperones contain one or more aromatics adjacent to an acidic stretch, and bind histones via a hydrophobic anchor and electrostatic helix capping mechanism. This "anchoring and capping" mechanism may be conserved in Npm and many other chaperones.

**Fig. 7** Functional regulation of Npm by intramolecular interactions. **a** Npm domain map and truncations used in these experiments. Truncation names and residue numbers shown on right. **b** Equilibrium tryptophan fluorescence binding curves derived from the Npm tail and truncations binding to either H2A/H2B or H3/H4. Points represent average values of three replicates for each binding curve. Standard error of the points is included during the calculation of the binding curves and dissociation constants. **c** Dissociation constants ($K_D$) and standard errors ($\pm$) of Npm tail and truncations derived from **b**. **d** Competitive pull-down assays using various truncations of the Npm tail and StrepII-tagged H2A/H2B dimers. Lanes 1–4, 10% inputs. Lanes 5–8, resin only controls. Lanes 9–12, standard pull-down assays. Lanes 13–18, competitive pull-down assays. Presence of Npm truncation bands in competitive pull-down lanes indicate higher affinity for H2A/H2B. **e** Chromatin assembly assay comparing the full-length Npm (residues 1–195) and tail domain (119–195). Concentrations of chaperone used in each indicated as molar ratio to histone octamer. **f** Chromatin assembly assay comparing full-length Npm (residues 1–195), ΔCterm10 (1–185), ΔCterm16 (1–179), and core+A2 (1–145) truncations at 11:1 molar ratio of Npm to histone octamer. **g** Quantification of chromatin assembly assays in **f**. 4–7 replicates of each shown with standard errors. One-way ANOVA and multiple group comparison of chromatin assembly activity of truncations to full-length Npm. ****$p < 0.0001$. **h** In vitro TTLL4 glutamylation assay using core+A2 truncation ± H2A/H2B. Glutamylation of Npm readout by anti-glutamylation western blot (top). Membrane stained in Direct Blue 71 (bottom). Histone binding inhibits Npm glutamylation on glutamates 127, 128, 129, and 131. **i** In vitro TTLL4 glutamylation assay using C-terminal truncations of Npm. Glutamylation of Npm readout by anti-glutamylation western blot (top). Membrane stained in Direct Blue 71 (bottom). The C-terminal 10 amino acids inhibit Npm glutamylation on glutamates 127, 128, 129, and 131. **j** Histone aggregate removal assay of linear DNA + H2A/H2B run on native TBE-PAGE with various truncations of Npm added at 2:1 molar ratio of Npm:H2A/H2B. Lower band indicates free DNA, upper bands indicate H2A/H2B:DNA aggregates. Core+A2 (1–145) is the only truncation capable of removing aggregates

Importantly, neither the tail domain nor A2 alone can deposit histones in the chromatin assembly assays nor remove H2A/H2B-DNA aggregates in the aggregation removal assays, whereas the core+A2 truncation has robust activity in both assays. These results indicate that both the tail domain and the core domain are necessary, but not sufficient, for proper histone shielding and deposition by Npm. This is counterintuitive given that we also showed that the GST-tail recapitulates the binding pattern of the full-length Npm and that the core domain shows very little binding in the histone peptide array. We conclude that despite having very weak affinity for histones[19], the core domain is necessary for the function of Npm as a histone chaperone[45].

Many structural models have been proposed to explain the histone binding activity of Npm. Previous SAXS and X-ray crystallography studies suggest that H2A/H2B bind to the lateral face of the pentamer, with A2 of the tail domain draped over the dimer forming electrostatic contacts necessary for charge shielding[16,19]. In contrast, negative-stain EM studies posit that H2A/H2B bind strictly to the distal face of the pentamer with minimal direct contacts being made with the core domain[17,46]. Our SAXS data of the core+A2:H2A/H2B complex, and modeling using NMR restraints suggest that H2A/H2B dimers bind to the upper portion of the lateral face around the pentamer. This positioning could enable favorable interactions between the first acidic stretch (A1) and H2A/H2B dimers, which is also in this region of the structure, as well as avoid clashes between adjacent dimers. The additional contacts observed between the Npm core domain and H2A/H2B in our SAXS/NMR hybrid models are likely crucial for the robust histone deposition and aggregate removal activities observed for the core+A2 truncation.

The short N-terminal 15 aa tail of Npm is also predicted to be disordered and is positioned directly adjacent to the longer C-terminal tail domain at the pentameric distal face. We previously showed that this short N-terminal tail is phosphorylated on multiple sites during development, which may also serve to regulate histone binding and acidic stretch accessibility of Npm[18]. Comparing the HSQC spectra from the tail domain with NpmFL, we observed that tails of NpmFL adopt similar, but slightly more open conformations. This could be an effect of the bulky core domain, or due to interactions between the C-terminal tail and the shorter N-terminal tail. Finally, from our SAXS positioning, it would not be surprising if this region also makes direct contacts with H2A/H2B dimers in the pentameric complex. Future studies into this region will be critical in determining its contribution to the histone binding and deposition activities of Npm.

The abundance of acidic stretches and disordered regions in histone chaperones and other chromatin binding proteins suggest that similar modes of regulation via acidic stretch shielding may be functionally relevant to many histone-interacting proteins[10].

## Methods

**Antibodies, peptides, and reagents**. All chemical reagents were obtained from Sigma, RPI, Toronto Research Chemicals, Cambridge Isotope Labs, or Fisher Scientific. Antibodies used in this study were the TβIII anti-monoglutamylation antibody[47], an anti-Npm core domain polyclonal antibody (Lampire Biological Laboratories)[18], and a commercially available anti-GST tag antibody (Pierce Cat# 8-326). Peptides derived from the Npm tail were obtained from JPT Peptide Technologies (Berlin, Germany). All histone peptides from the microarray were synthesized and coupled to the chip by JPT.

**Protein expression, labeling, and purification**. Full-length recombinant Npm, as well as truncations retaining the core domain, was expressed in bacteria with a TEV cleavable His$_6$ tag and purified as described previously[18]. The tail domain, as well as truncations not retaining the core domain, was expressed in bacteria with a TEV cleavable His$_6$+GST tag to promote solubility, and purified by Ni$^{2+}$ chromatography. Cleavage of the tag was followed by subtractive Ni$^{2+}$ chromatography and cation-exchange chromatography on a Mono-S column to purify the tail domain. The tail domain was concentrated in a 3 kDa MWCO concentrator (Millipore).

Recombinant histones were purified as described previously[48]. Briefly, individual *Xenopus* histones were expressed in bacteria with a TEV cleavable His$_6$ tag, and purified from inclusion bodies by Ni$^{2+}$ chromatography under denaturing conditions. After tag cleavage and subtractive Ni$^{2+}$ chromatography, H2A and H2B or H3 and H4 were mixed at stoichiometric ratios under denaturing conditions, and refolded into dimers or tetramers by stepwise dialysis against a non-denaturing high salt buffer. Dimers and tetramers were further purified by size-exclusion chromatography using a Superdex75 column. Fractions containing the H2A/H2B dimers or H3/H4 tetramers were then dialyzed into a low-salt buffer and concentrated in a 10 kDa MWCO concentrator (Millipore).

For isotope labeling, 4 L bacterial cultures were grown in LB media to an OD$_{600}$ of approximately 0.7. Bacteria were pelleted by centrifugation, washed with 1× M9 salts (carbon and nitrogen free), and transferred to 1 L of M9 minimal media supplemented with either 1 g/L of $^{15}$N ammonium chloride (For $^{15}$N labeling) or 1 g/L of $^{15}$N ammonium chloride + 2 g/L of $^{13}$C D-glucose (For $^{13}$C, $^{15}$N labeling). Bacteria were shaken for 1 h at 37 °C to allow clearing of unlabeled metabolites prior to induction with 0.5 mM IPTG for 3–4 h at 37 °C[49]. Isotope-labeled tail was purified as described above.

Paramagnetic spin labeling of cysteine mutants of the tail was performed in high salt buffer (25 mM Na$_2$PO$_4$, pH 7.0, 1 M NaCl, and 1 mM EDTA) after purification. Briefly, cysteine residues in the tail were reduced by addition of 10 mM DTT at room temperature for 30 min. The samples were then concentrated and serially buffer exchanged to high salt buffer without DTT using a 3 kDa MWCO concentrator until final DTT concentration was <0.02 mM. A 10× molar excess of MTSL (Toronto Research Chemicals) from a 100 mM stock in methanol was added to the samples and incubated overnight at 4 °C protected from light. The samples were then serially dialyzed into either low-salt (150 mM) or high salt (1 M) buffer until final free MTSL concentration was <1 nM. Spin labeling of H2A or H2B was performed similarly, except under denaturing conditions (20 mM Tris, pH 7.6, 25 mM NaCl, 6 M Guanidine-HCl). Dimers were then refolded by serial dialysis and purified by size-exclusion chromatography as described above[48]. Labeling efficiency was estimated using Ellman's reagent for the detection of unlabeled cysteine residues, using Cys-HCl to generate a standard curve. Labeling efficiency was consistently measured at >95%.

**Nuclear magnetic resonance spectroscopy**. Sample preparation: Samples of the Npm tail were prepared in 25 mM Na$_2$PO$_4$, pH 7.0, 150 mM NaCl, 1 mM EDTA, 10% D$_2$O, and 0.05 mM TSP as a reference (NMR buffer). For resonance assignment $^{13}$C, $^{15}$N-labeled tail was concentrated to 0.86 mM in NMR buffer, except NaCl was lowered to 25 mM. This did not result in significant chemical shift changes compared to 150 mM NaCl condition. NaCl titrations were performed using 0.3 mM of $^{15}$N-labeled tail with subsequent additions of NMR buffer+4 M NaCl to the sample and concentrating back to the original volume. H2A/H2B titrations were performed using 0.3 mM of $^{15}$N-labeled tail by addition of a concentrated stock of recombinant H2A/H2B dimers in NMR buffer to the sample and concentrating back to the original volume. H3/H4 titration was performed in the same manner, except the tail domain was kept at 0.1 mM (aggregation occurred with H3/H4 at higher concentrations). For all relaxation measurements (T$_1$, T$_2$, and $^1$H-$^{15}$N NOE) the tail domain was kept at 0.5 mM in NMR buffer. For PRE experiments, the tail was kept at either 0.2 mM (unbound) or 0.4 mM (H2A/H2B bound) in either NMR buffer (low salt), or NMR buffer with 1 M NaCl (high salt). Spin labels were reduced by addition of 2 mM ascorbic acid pH 7.0 from a 500 mM stock (<0.5% sample dilution) and incubated for 30 min at room temperature prior to data collection. Expected peak positions in the HSQC spectra of spin-labeled Npm tail in the apo and histone-bound states confirm that the addition of spin labels did not significantly alter tail conformation or histone-binding properties.

Data collection, processing, and analysis: All NMR data were collected at 298 ˚K on either a Bruker DRX600 or Varian INOVA600 spectrometer equipped with a 5 mm cryo-probe. Data were processed in either Bruker Topspin or NMRPIPE software[50] and analysis was carried out in CcpNMR analysis software[51]. Typical $^1$H-$^{15}$N HSQC spectra were collected using between 8–16 scans, 2048 × 128 data points and using 12 × 28 ppm spectral windows. For assignment, a standard suite of triple resonance experiments was carried out on the $^{13}$C, $^{15}$N-labeled Npm tail domain (HNCA, HNCACB, HNCO, HN(CO)CA, CBCA(CO)NH, HN(CA)CO, and HBHA(CO)NH). Resonances were assigned using a combination of PINE NMR server predictions[52] and manual inspection of the spectra. CSPs upon NaCl and histone titration were calculated using the following equation:

$$\text{CSP} = \sqrt{(\Delta\delta_{\text{HN}})^2 + (\Delta\delta_N * \alpha_N)^2}, \tag{1}$$

Where $\Delta\delta_{\text{HN}}$ is the change in chemical shift in the proton dimension, $\Delta\delta_N$ is the change in chemical shift in the nitrogen dimension, and $\alpha_N$ is a scaling factor of 0.14.

For PRE measurements, two $^1$H-$^{15}$N HSQC spectra were acquired on the same sample prior to and after reduction of the spin label. Peak intensities were measured, and plotted as the ratio of intensities in the oxidized state ($I_{\text{ox}}$) to intensities in the reduced state ($I_{\text{red}}$). Standard errors in $I_{\text{ox}}/I_{\text{red}}$ values were estimated from propagation of the signal-to-noise ratio of the individual spectra

using the following standard equation for error propagation:

$$\frac{\delta I_{ox}/I_{red}}{I_{ox}/I_{red}} = \sqrt{\left(\frac{\delta I_{ox}}{I_{ox}}\right)^2 + \left(\frac{\delta I_{red}}{I_{red}}\right)^2}, \qquad (2)$$

Where $\delta I_{ox}/I_{red}$ is the calculated error in the $I_{ox}/I_{red}$ ratio, $\delta I_{ox}$ is the noise level of the oxidized spectrum, and $\delta I_{red}$ is the noise level of the reduced spectrum. Residues with $\delta I_{ox}/I_{red} \geq 0.3$ were excluded from the graphs and all subsequent analyses.

$T_1$ and $T_2$ measurements were performed by using inversion-recovery and spin-echo sequences incorporated into $^1$H-$^{15}$N HSQC pulse sequences and using 10 variable delay times (10–1800 ms for $T_1$, 0–1017 ms for $T_2$). The $T_2$ experiment employed a CPMG pulse train that includes $^1$H 180 degree pulses to eliminate the effects of cross-correlation between $^1$H-$^{15}$N dipolar coupling and chemical shift anisotropy relaxation mechanisms and a delay of 0.9 ms is inserted between successive applications of $^{15}$N 180 degree pulses in the CPMG pulse train. Peak intensities of each assigned residue were measured as a function of delay time, and fit to a single exponential decay to extract $T_1$ and $T_2$ values, as well as standard error from the fits. To measure $^1$H-$^{15}$N heteronuclear NOE ratios, a reverse 2D INEPT experiment was performed with and without $^1$H saturation in an interleaved manner. Peak intensities for each residue were measured with and without $^1$H saturation, and the data were plotted at the ratio of the two values (saturated to unsaturated). Standard errors in NOE ratios were estimated from propagation of the signal-to-noise ratio of the individual spectra.

**PRE-based structural modeling**. The paramagnetic contribution to the spin relaxation ($R2^{sp}$) is the difference in transverse relaxation rates between the paramagnetic (oxidized) and diamagnetic (reduced) samples:

$$R2^{sp} = R2^{paramagnetic} - R2^{diamagnetic}. \qquad (3)$$

To structurally model the ensemble of the tail domain in the unbound state, we first estimated $R2^{sp}$ from the measured intensity ratios using the following equation:

$$\frac{I_{ox}}{I_{red}} = \frac{R2 \exp(-R2^{sp}t)}{(R2 + R2^{sp})}, \qquad (4)$$

Where $\frac{I_{ox}}{I_{red}}$ is the measured intensity ratio, R2 is the spin relaxation rate of the diamagnetic sample, and $t$ is the scan time (16.6 ms). We calculated R2 in the diamagnetic sample by measuring peak widths at half-height using NMRPIPE autoFit function. We then generated an extended model of the tail using FlexibleMeccano software[53]. We used Xplor-NIH software to replace native residues G132, T155, S172 with Cys-MTSL and added a Cys-MTSL residue onto the extreme C-terminus[54]. We used 140 calculated intramolecular PRE restraints from the four spin label positions, and estimated the error in $R2^{sp}$ based on the errors in $I_{ox}/I_{red}$ ratios. $I_{ox}/I_{red}$ ratios with errors $\geq 0.3$ were considered too noisy and excluded from structure calculations. Errors in calculated $R2^{sp}$ values were used as weighting factors during the ensemble simulations, with those $R2^{sp}$ values with smaller errors weighted more heavily than those with larger errors during the simulated annealing. $R2^{sp}$ values with errors $< 1.0 \text{ s}^{-1}$ were set to a minimum value of 1.0 to avoid overweighting these PRE effects beyond the accuracy of the measurement. We performed simulated annealing ensemble structure calculations using the PREPot term and an average-type PRE-target function in Xplor-NIH similar to that described previously[55,56]. After an initial minimization, the conformers were heated to 3000 K, and slowly cooled to 100 K in 25 K increments, while ramping the PRE force constant across the ensemble. Other forces included during the simulated annealing were the van der Waals repulsive term and covalent energy terms. 50 ensembles were calculated this way using a specified number of conformers (1–10 conformers per ensemble). Agreement between each ensemble and the experimental PRE data was assessed by Q-factor analysis:

$$Q = \sqrt{\frac{\sum_i \{R2sp^{obs}(i) - R2sp^{calc}(i)\}^2}{\sum_i R2sp^{obs}(i)^2}}, \qquad (5)$$

Where R2sp$^{obs}$ are the experimentally observed $R2^{sp}$ values and R2sp$^{calc}$ are the back-calculated $R2^{sp}$ values from the ensemble. Ensembles were sorted by Q-factor, and the top five in each group (lowest Q-factor) were compared and used for further analysis.

For the histone-bound structures, we imported the H2A/H2B structure from the nucleosome structure (PDB: 1AOI). We replaced native residues H2A E64, H2B S61, H2B G72, H2C T116 with Cys-MTSL residues in Xplor-NIH. We also imported a FlexibleMeccano model of residues 119–149 (A2) of the Npm tail domain. We calculated $R2^{sp}$ values from $I_{ox}/I_{red}$ ratios in the bound state, as in Eq. 4, and then applied calculated $R2^{sp}$ values to the following equation to calculate an average interatomic distance between the MTSL nitroxide group and amide protons of A2 (r):

$$r = \left[\frac{K}{R2^{sp}} \left(4\tau_c + \frac{3\tau_c}{1 + \omega_h^2 \tau_c^2}\right)\right]^{\frac{1}{6}}, \qquad (6)$$

Where K is a constant related to the spin properties of the system ($K = 1/15 \times S(S+1)\Upsilon^2g^2\beta^2 = 1.23 \times 10^{-32} \text{ cm}^6 \text{ s}^{-2}$), $\tau_c$ is the protein correlation time, and $\omega_h$ is the larmor frequency for the amide proton. We estimated $\tau_c$ of the A2:H2A/H2B complex based on molecular weight and calculations by HydroNMR software.

During the structure calculations we fixed the histone fold of the H2A/H2B dimers to allow docking of A2, while allowing the histone tails to remain flexible. We applied 43 intermolecular distance restraints from the four spin label positions on H2A/H2B to amide protons of A2. We allowed a $\pm 2$ Å error on calculated distances from $I_{ox}/I_{red}$ ratios between 0.15 and 0.8. For $I_{ox}/I_{red}$ ratios $< 0.15$ (strong PRE effects), we allowed the distance to vary from 0 to the calculated distance $+2$ Å. For $I_{ox}/I_{red}$ values $> 0.8$ (weak PRE effects), we allowed the distance to vary from calculated distance $-2$–70 Å. Simulated annealing was performed in a similar manner to the unbound tail. 1000 conformers of A2:H2A/H2B were calculated this way. These structures were sorted by energy, and the top 100 structures were aligned in PyMOL software. To better visualize the most occupied positions of A2 within the large ensemble of structures, we used Xplor-NIH to generate a reweighted atomic probability density map of A2 around the H2A/H2B dimer, similar to that described previously[57]. Density maps were contoured and imaged in UCSF Chimera software[58].

**Molecular dynamics**. The coordinates of the Npm tail domain (residues 119–195) in an extended conformation were generated using FlexibleMeccano software. Na$^+$ counter ions were added to neutralize the protein charge, and the protein was solvated in a box of TIP4P-D water with a distance of 20 Å between the protein surface and the wall of the box[29]. Simulations were performed in AMBER using the AMBER99SB-ILDN force field, and periodic boundary conditions were imposed[59,60]. The box was first minimized for 250 steps of steepest descent minimization, followed by 250 steps of conjugate gradient minimization, with the protein atoms fixed with a 500 kcal $^{-1}$mol $^{-1}$ Å$^2$ restraint. Next, the box was heated to 300 K over 20 ps using a Langevin thermostat, and NPT equilibration was performed for 200 ps; these steps were performed with 10 kcal mol$^{-1}$ Å$^2$ restraints on the protein atoms. This was followed by another 2 ns of NVT equilibration, before beginning production runs. For all simulations, a 12 Å cutoff was used. Long-range electrostatics were handled using the Particle mesh Ewald algorithm[61]. Hydrogen bonds were constrained using the SHAKE algorithm. The time step for all simulations was 2 fs.

**Circular dichroism**. Circular dichroism was performed in a Jasco spectro-photometer using a quartz cuvette with a 0.1 cm path length. The Npm tail was read at a concentration of 0.2 mg mL$^{-1}$ in 10 mM Na$_2$PO$_4$ pH 7.0, 10 mM NaCl, and 1 mM EDTA. Scans were performed in triplicate at 25 °C and data were measured in 0.1 nm steps between 250 and 190 nm. A buffer only background correction was subtracted from the data, the data were converted from milli degrees to mean residue ellipticity, [$\theta$], and secondary structure was estimated by CONTIN analysis.

**Analytical ultracentrifugation**. AUC sedimentation velocity experiments were performed on a Beckman XL-I Ultracentrifuge equipped with a Ti-60 rotor using the absorption optics at 280 nm. The Npm tail domain was run at 58,000 rpm (345,496 RCF) at three concentrations (15, 50, and 100 μM) in 25 mM Na$_2$PO$_4$ pH 7.0, 1 mM EDTA and various NaCl concentrations at 20 °C. Sednterp version 20120828 beta was used to calculate the partial specific volume of the proteins from their sequence and the density and viscosity of the buffers. The sedimentation parameters were corrected to standard conditions (20,w) using these values. Seventy scans were collected over the course of a run. A subset of scans, beginning with those where a clear plateau was evident between the meniscus and the boundary, were selected for time-derivative analysis using DCDT+ver. 2.4.2.

**Chromatin assembly assays**. Chromatin assembly assays were performed as described previously[18]. Briefly, 350 ng of hyperacetylated HeLa core histones were premixed with Npm at the indicated molar ratio for 20 min on ice in 20 mM HEPES pH 7.5, 60 mM KCl, 2 mg mL$^{-1}$ BSA, and 1.2% PEG8000 and polyvinyl alcohol. 350 ng of pGIEO plasmid DNA was relaxed by incubation with the topoisomerase ND423 for 5 min at 27 °C in a separate reaction containing 50 mM Tris pH 7.5, 10 mM MgCl$_2$, 0.1 mM EDTA, 50 μg mL$^{-1}$ BSA, and 0.5 mM DTT. The two solutions were mixed along with 280 ng of the ISWI ATP-dependent chromatin remodeling enzyme and an ATP regeneration system containing 20 mM HEPES pH 7.5, 2 mM ATP, 2 mM MgCl$_2$, 10 μg mL$^{-1}$ creatine kinase, 10 mM creatine phosphate, and 1 mM DTT. This reaction was incubated for 2 h at 27 °C to allow nucleosome assembly to occur on the plasmid DNA. The reactions were stopped and proteins were digested by addition of a buffer containing 10 mM Tris pH 8.0, 1% SDS, 1 mM EDTA, 75 μg mL$^{-1}$ GlycoBlue, and 0.25 mg mL$^{-1}$ Protei-nase K and heating to 60 °C for 1 h. The plasmid DNA was purified by phenol–chloroform extraction and ethanol precipitation. The purified plasmid DNA was then slowly resolved at 25 V on a 1.2% agarose gel for 16 h at 4 °C to separate differentially supercoiled forms. The gel was post-stained in a 2 μg mL$^{-1}$ ethidium bromide solution to visualize the DNA. Percent-supercoiled DNA was quantified using ImageQuant software by taking the ratio of intensities of the lowest band (fully supercoiled) to the sum of the entire DNA in the lane. Percent-

supercoiled DNA was then normalized to the highest supercoiled DNA on the same gel to generate the supercoiling index metric.

**In vitro glutamylation assays**. In vitro glutamylation assays of Npm truncations were performed as described previously[18]. Briefly, 2 µg of Npm was mixed with 5 µg of recombinant GST-tagged *Xenopus* TTLL4 catalytic domain (GST-TTLL4Δ526) in the buffer: 25 mM Tris pH 8.6, 10 mM MgCl$_2$, 2.5 mM DTT, 0.1 mM L-glutamate, and the ATP regeneration system (see "Chromatin Assembly Assay" in Methods section). Reactions were incubated at 30 °C overnight. Half of the reaction (1 µg of Npm substrate) was resolved by 15% SDS–PAGE, and transferred to a PVDF membrane for western blot detection by the rabbit TβIII anti-glutamylation primary antibody (1:10,000 dilution) and a HRP α-rabbit secondary antibody (1:100,000 dilution). In Fig. 7h, recombinant H2A/H2B dimers were premixed at 1:1 stoichiometry with the chaperone prior to adding the enzyme, and control samples were buffer matched.

**Histone aggregate removal assays**. A ~500 bp linear, random DNA sequence in 25 mM Na$_2$PO$_4$ pH 7.0, 150 mM NaCl, and 1 mM EDTA at 15.3 nM was incubated with H2A/H2B dimers at a 10× molar excess (153 nM) for 15 min at 23 °C to form a pattern of aggregates observable by native TBE-PAGE. A 2× (306 nM) or 10× (1530 nM) molar excess of Npm or truncations were added, and incubated for an additional 30 min at 23 °C. 1.2 µL of 50% glycerol was added to 10 µL of the reaction and the reaction was loaded on a 5% native TBE-PAGE gel. The gel was run at 150 V for 45 min at 4 °C. The gel was then post-stained in a 2 µg mL$^{-1}$ ethidium bromide solution and imaged. Aggregation removal was observed by loss of aggregate bands and increased intensity of the free DNA band.

**Competitive pull-down assays**. H2A with a C-terminal StrepII tag was used to form dimers with untagged H2B as described above[48]. 10 µg of StrepII-tagged H2A/H2B dimers was coupled to Streptactin superflow resin in binding/wash (BW) buffer (100 mM Tris-HCl pH 7.6, 150 mM NaCl, 1 mM EDTA, 5 mM BME) for 2 h at 4 °C. Resin was washed and incubated with 5 µM of each truncation, either alone or mixed, in 100 µL volume overnight at 4 °C. The resin was extensively washed in BW buffer and resuspended in 1× Laemmli buffer. ~20% of the pull-down reaction was separated by 15% SDS–PAGE and stained in Coomassie Brilliant Blue. Resin only controls were performed in the same manner with buffer substituted for StrepII-tagged H2A/H2B.

**Tryptophan fluorescence binding assays**. Tryptophan 125 fluorescence of the Npm tail domain in 20 mM Tris pH 7.6, 150 mM NaCl, 1 mM EDTA, 1 mM DTT, and 0.01% NP-40 was measured in a Fluoromax 3 spectrofluorometer. The tail domain at 0.5 µM in a 0.1 × 0.1 cm quartz cuvette was excited at 295 nm using 5 × 5 nm slits. Buffer subtracted emission spectra were collected in 1 nm steps between 305 and 450 nm using a 0.5 s integration time per step. H2A/H2B or H3/H4 were titrated in the same buffer, and background autofluorescence from the histones was subtracted by reading identical samples without the tail domain. The emission peak of the tail alone was consistently centered at 358 nm with a 340/360 intensity ratio of ~0.8. Addition of both H2A/H2B and H3/H4 lead to a shift in the emission spectra toward lower wavelengths (blue shift). The data were initially plotted as the change in intensity at 340 nm/intensity at 360 nm (Δ340/360), where binding to H2A/H2B lead to a maximum shift of ~0.33 and binding to H3/H4 lead to a maximum shift of ~0.16. Curves were converted to fractional saturation (Y) by normalizing Δ340/360$_{min}$ and Δ340/360$_{max}$ values to 0 and 1, respectively. Equilibrium dissociation constants (K$_D$) and standard errors were then calculated by fitting the data to a single-site binding model taking the concentration of the tail into account:

$$Y = \frac{(P + L + K_D) - \sqrt{(P + L + K_D)^2 - 4PL}}{2P}, \quad (7)$$

Where Y is the normalized response (0–1), P is the concentration of the tail domain (0.5 µM), L is the concentration of the histone ligand, and K$_D$ is the equilibrium dissociation constant. Using this equation and tail domain concentration, we are able to accurately measure dissociation constants > 50 nM with little error. Stoichiometries of the complexes were confirmed by titrating histones against 10 µM of the tail domain in similar experiments, and fitting Δ340/360 values to a segmental linear regression, where the breakpoint indicates stoichiometry of the complex (Supplementary Fig. 7c).

**Histone peptide array**. Histone peptide array experiments were performed by JPT Peptide Technologies (Berlin, Germany) (Cat# His_MA_01). Binding events were probed by using either a rabbit antibody raised against the Npm core domain (for detection of the full-length Npm and core truncation) or a rabbit α GST antibody (for detection of the GST-tail domain). Fluorescently labeled α rabbit IgG was used for detection of binding events. Signal intensities were scaled to the maximum of each primary antibody (i.e., full-length Npm and core truncations were scaled together, GST-tail was scaled independently). Binding events were mapped onto

both a linear sequence of the histones, as well as the crystal structure of the nucleosome (PDB: 1AOI).

**Small angle X-ray scattering and modeling**. Online size-exclusion chromatography small-angle X-ray scattering (SEC-SAXS) was performed at SSRL (Stanford Synchrotron Radiation Lightsource) Bio-SAXS beamline 4–2 in a similar manner as previously reported[62–65]. Experimental setup and parameters are summarized in Supplementary Table 1. 90 µL of the complex (Npm core+A2 bound to five H2A/H2B dimers) at 4.83 mg mL$^{-1}$ was applied to a Superdex 200 PC3.2/300 column (GE Healthcare, Wisconsin, USA). 5 mM dithiothreitol was added to both sample and buffer solution to limit radiation damage. 600 images were recorded with 1 s exposure every 5 s at 0.05 mL min$^{-1}$ flow rate. The program SasTool (http://ssrl.slac.stanford.edu/~saxs/analysis/sastool.htm) was employed for data reduction including scaling, Azimuthal integration, averaging, and background subtraction. The first 100 images at the early part of the void volume were averaged and used as a buffer-scattering profile for the background subtraction.

The script *fplcplots*, available at SSRL beamline 4–2, was used for consecutive Guinier analysis, implemented in the program AUTORG[66], and assessing data quality (e.g., radiation damage and cleanness of sample cell) (Supplementary Fig. 7). Since marginal inter-particle interactions (concentration dependence) were observed over the peak, two average profiles (image number: 430–479 and 470–484) were generated, scaled and merged for further analyses. Kratky plot of the merged scattering profile shows a prominent "bell-shape" peak at low q range and is well converged to a baseline at high q range, indicating that sample is globular without significant flexibility. Pairwise distribution function, P(r), was calculated using the program GNOM[67].

The program CORAL was employed for SAXS-based rigid body modeling[66]. Residues 15–118 of chain B in PDB ID:1K5J was used for a rigid model of Npm core. Prior to the modeling, a short disordered region corresponding to the first acidic stretch (residues 33–40) was reconstructed by the program MODELLER[68]. N-terminal residues 1–14 lacking from the crystal structure were reconstructed by CORAL. The A2 region of the top NMR structures (residues 124–149) was used as a rigid model. Residues 119–123 were allowed to remain as a flexible linker between the core domain and the reminder of A2 during the modeling. Residues 20–103 of chain C and 33–119 of chain D in PDB:1KX5 were used for individual rigid models of H2A and H2B, respectively. N- and C-terminal regions of H2A and H2B were assigned as lacking fragments to be reconstructed. Based on the NMR structures, H2A and H2B positions were fixed along with A2 and those three components (A2: H2A/H2B) were moved together around Npm core domain during the modeling. The coordinates of the Npm core domain (residues 15–118 of B chain) was reoriented in the z-axis so that identical pentameric interfaces observed in the crystal structure of Npm core were reproduced by a five-fold rotational operation. The Npm core domain was then fixed and P5 symmetry was applied during the modeling.

30 independent ab initio models were generated using DAMMIF with P5 symmetry and the anisometry option of oblate shape[69]. The resulting structures were averaged and filtered using the program DAMAVER (Supplementary Table 1)[70].

**Data availability**. Assigned chemical shifts for the Npm tail domain have been deposited in the Biological Magnetic Resonance Data Bank (BMRB) under the deposition code: 26809. SAXS data and envelope have been deposited at the Small Angle Scattering Biological Data Bank (SASBDB) under the deposition code: SASDBY4.

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

## Acknowledgements

We thank T. Owen-Hughes and M. Shrogren-Knaak for providing histone cysteine point mutant constructs and W.L. Wang for StrepII-tagged H2A/H2B dimers. We thank C. Schwieters and L. Deshmukh for helpful discussions on PRE-based structural modeling. This work was supported by The American Cancer Society-Robbie Sue Mudd Kidney Cancer Research Scholar Grant (124891-RSG-13-396-01-DMC) and NIH grant R01GM108646 (both to D.S.) and training grants T32GM007491 and F31GM116536 to C.W. J.M.K. was supported by the Einstein MSTP Training Grant (T32 GM007288). The Bruker 600 NMR instrument was purchased using funds from NIH award 1S10OD016305 and is supported by the Albert Einstein College of Medicine. The Inova 600 NMR instrument in the Einstein Structural NMR Resource was purchased using funds from NIH award 1S10RR017998 and NSF award DBI0331934 and is supported by the Albert Einstein College of Medicine. Use of the Stanford Synchrotron Radiation Lightsource, SLAC National Accelerator Laboratory, is supported by the U.S. Department of Energy, Office of Science, Office of Basic Energy Sciences under Contract No. DE-AC02-76SF00515. The SSRL Structural Molecular Biology Program is supported by the DOE Office of Biological and Environmental Research, and by the National Institutes of Health, National Institute of General Medical Sciences (including P41GM103393). The contents of this publication are solely the responsibility of the authors and do not necessarily represent the official views of NIGMS or NIH.

## Author contributions

C.W. conceived and performed experiments, interpreted results, and wrote the manuscript. T.M. performed SAXS experiments, interpreted results, built models, and wrote the manuscript. J.M.K. and D.C. aided in the set up and running of molecular dynamics simulations, interpreted results, and wrote the manuscript. T.O. conceived experiments and provided Npm truncation and histone constructs. S.C. aided in setup and S.C. and D.C. aided in analysis of NMR experiments. M.B. conceived, conducted, and interpreted the AUC analysis and edited the manuscript. M.G. conceived and interpreted NMR experiments, performed structure calculations, and edited the manuscript. D.S. conceived and interpreted experiments, wrote the manuscript, and supervised all work. All authors read and approved the final manuscript.

## Additional information

**Competing interests:** The authors declare no competing financial interests.

