## [Peer Review File · Nature Communications]

REVIEWERS' COMMENTS:

Reviewer #2 (Remarks to the Author):

In this revised manuscript, Warren et al. provide an updated study of the intrinsically disordered region of the nucleoplasmin (Npm) histone chaperone. The modeling of IDPs is notoriously difficult. In contrast to well folded proteins, where a static snapshot can often provide an excellent model of the average conformational state of the protein, it is very challenging to build such a model of the conformation of IDPs, which sample a much broader conformational landscape. However, the importance of these proteins and protein regions in many biological processes makes it critical to tackle the challenge of defining their structural properties. Many concerns were raised previously about the derived model of the Npm Tail domain and its interaction with the H2A/H2B dimer. In this revision, the authors have added additional data and altered the manner in which they are generating the final model of the apo Tail domain and the complex. Though I agree with the other reviewers that a single model will not satisfactorily represent the full conformational ensemble I do not think that the authors attempt to make any such claim. The way in which the data is analyzed and discussed clearly represents that a broad conformational distribution is available to the Tail domain both in the apo form and in complex with the H2A/H2B dimer. However, the authors are also able to satisfactorily characterize the contacts that drive the Tail domain intramolecular interactions and those that drive complex formation. This model is further satisfactorily validated in the binding and functional assays carried out.

All of my previous concerns have been addressed in this most recent version of the manuscript and I believe this manuscript is appropriate for publication in Nature Communications. My only additional comment is that in the discussion of the functional assays presented there is a lack of presented rationale. Though the results are convincing it would be nice to present to the reader why, for instance, glutamylation assays are being carried out.

Referee #2:

In this revised manuscript, Warren et al. provide an updated study of the intrinsically disordered region of the nucleoplasmin (Npm) histone chaperone. The modeling of IDPs is notoriously difficult. In contrast to well folded proteins, where a static snapshot can often provide an excellent model of the average conformational state of the protein, it is very challenging to build such a model of the conformation of IDPs, which sample a much broader conformational landscape. However, the importance of these proteins and protein regions in many biological processes makes it critical to tackle the challenge of defining their structural properties. Many concerns were raised previously about the derived model of the Npm Tail domain and its interaction with the H2A/H2B dimer. In this revision, the authors have added additional data and altered the manner in which they are generating the final model of the apo Tail domain and the complex. Though I agree with the other reviewers that a single model will not satisfactorily represent the full conformational ensemble I do not think that the authors attempt to make any such claim. The way in which the data is analyzed and discussed clearly represents that a broad conformational distribution is available to the Tail domain both in the apo form and in complex with the H2A/H2B dimer. However, the authors are also able to satisfactorily characterize the contacts that drive the Tail domain intramolecular interactions and those that drive complex formation. This model is further satisfactorily validated in the binding and functional assays carried out.

We thank the reviewer for these supportive comments and recognition of the extensive and complementary tests we performed to test our hypothesis.

All of my previous concerns have been addressed in this most recent version of the manuscript and I believe this manuscript is appropriate for publication in Nature Communications. My only additional comment is that in the discussion of the functional assays presented there is a lack of presented rationale. Though the results are convincing it would be nice to present to the reader why, for instance, glutamylation assays are being carried out.

We ensured that the rationale for the functional tests was clear in the text. We revised the glutamylation paragraph by moving the rationale to the first sentence, as follows: “We next tested these C-terminal truncations for their ability to be glutamylated by the catalytic domain of the enzyme tubulin tyrosine ligase-like 4 (TTL4Δ526), as this assay is an effective readout of A2 solvent accessibility.”